# The Relationship between Resistance Training Frequency and Muscle Quality in Adolescents

**DOI:** 10.3390/ijerph19138099

**Published:** 2022-07-01

**Authors:** Marshall A. Naimo, Ja K. Gu

**Affiliations:** 1Military Performance Division, United States Army Research Institute of Environmental Medicine, Natick, MA 01760, USA; 2Centers for Disease Control and Prevention, National Institute for Occupational Safety and Health, Health Effects Laboratory Division, Morgantown, WV 26505, USA; jgu@cdc.gov

**Keywords:** muscle composition, exercise prescription, minimum dose, adolescence, cohort study

## Abstract

Previous research has established the role of resistance training (RT) on muscle function in adolescents, but a lack of evidence to optimize RT for enhancing muscle quality (MQ) exists. This study examined whether RT frequency is associated with MQ in a nationally representative adolescent cohort. A total of 605 adolescents (12–15 year) in NHANES were stratified based on RT frequency. MQ was calculated as combined handgrip strength divided by arm lean mass (via dual-energy X-ray absorptiometry). Analysis of covariance was adjusted for sex, race/ethnicity, and arm fat percentage; *p* < 0.05 was considered significant. RT frequency was associated with MQ for 2–7 day/week but not 1 day/week. When no RT was compared to 1–2 and 3–7 day/week, associations were present for 3–7 day/week but not 1–2 day/week. When comparing no RT to 1–4 and 5–7 day/week, associations existed for 5–7 day/week but not 1–4 day/week. Next, no RT was compared to 1, 2–3, and 4–7 day/week; associations were found for 4–7 day/week, while 2–3 day/week had a borderline association (*p* = 0.06); there were no associations for 1 day/week. Finally, no RT was compared to 1, 2, 3, 4, and 5–7 day/week; associations were present for all except 1 and 3 day/week. These prospective data suggest a minimum RT frequency of 2 day/week is associated with MQ in adolescents as indicated by the lack of differences in MQ between 1 day/week RT versus no RT.

## 1. Introduction

The capability of skeletal muscle to produce force in short periods of time is crucial in many activities of daily living as well as physically demanding tasks [1]. In applied and clinical research settings, muscle quality (MQ) is often quantified as muscle strength or power per unit of muscle mass [2,3], which accounts for the functional significance of the skeletal muscle tissue, and ultimately, its capabilities to produce force [4]. Increased force output from increased muscle size is a well-established physiological concept [5]; however, research demonstrates that force production stems from MQ, aside from the overall amount of muscle [6,7,8,9]. In addition to positive improvements in muscle strength/size, research has shown resistance training (RT) has numerous beneficial effects for adolescents on several factors that would subsequently improve MQ, such as improved functional motor performance, reduced central and whole-body adiposity, increased muscle activation rates, increased tendon CSA and stiffness, and improved bone strength [10,11,12].

The distinction between MQ and muscle quantity is an important aspect of functional assessment because individuals may have similar amounts of muscle mass but not necessarily equivalent strength [1]. For instance, on the surface it appears that obese adolescents have greater maximum strength compared to non-obese adolescents, suggesting that higher bodyweight as a consequence to greater adipose tissue may serve as a chronic stimulus on the leg muscles that aids in increasing muscle size and strength [13]. However, when strength is normalized to body mass, relative strength to body mass is less in obese individuals compared to their non-obese counterparts [13]. Interestingly, when strength is measured relative to muscle mass (i.e., MQ), current evidence suggests there is no effect of obesity on MQ, although potential differences in methods employed (e.g., control of recreational physical activity levels between groups, techniques employed to measure muscle mass, etc.) may account for the lack of differences seen [13]. 

Previous research in both humans and animals has established the role of RT on strength, function and MQ in adolescents [14,15,16,17,18]. Studies report that the response to RT may be optimal during adolescence and at late adolescence transitioning into early adulthood [18,19], although a lack of muscle adaptation in response to RT has also been demonstrated [20]. Moreover, animal studies using a physiologically representative model of RT have shown that 3-month-old rats, an age considered to be in late adolescence/early adulthood, have reported increases in muscle strength and size in the exercised limb following training in comparison to adult and old rats [16,21,22] in comparison to adult and old rats. However, muscle weights in non-loaded control muscles of 3-month-old adolescent rats have been found to be no different from those of 30-month-old rats, which was attributed to the 3-month-old rats not having reached their full skeletal maturation [22]. 

Although physiological adaptations to RT in adolescents follow the same guiding principles as adults, the adaptive response to RT is potentially different in this age group and may be more understated [10]. Potential factors that may lead to differential responses to RT in adolescents include large increases in circulating hormones (e.g., androgens) and various growth factors during puberty, which aids in accelerating growth and development of muscle tissue during this period of time, and hence can potentially hide effects seen in the muscle adaptive response during RT because it is hard to distinguish the influences of pubertal growth versus training-induced adaptations [10]. It has also been stated that adolescents have larger initial strength improvements to RT compared to adults that is attributed to heightened neural adaptations as a result of a physiological maturation of the central nervous system [11]. Given the unique and differential responses of adolescents to RT, a further characterization of the systematic RT exercise prescription that is appropriate for inducing changes in muscle adaptation and MQ in this specific age cohort. 

Training frequency is an often overlooked variable of exercise prescription [23]. While studies and systematic reviews have been conducted on RT frequency [24,25,26,27,28], current recommendations for improving muscular adaptation are based on inferences from a limited body of evidence with inherent major limitations that makes the interpretation and applicability of results difficult to implement, such as small sample sizes [26,27,29]. Moreover, none of these studies reported changes in MQ, and a clear minimal dose of RT in adolescent individuals is unknown. Moreover, there is a clear lack of evidence on how to optimize programs for power, strength, and hypertrophy necessary to enhance MQ in adolescents.

To the best of our knowledge, no studies have examined the relationship between RT and MQ in a nationally representative cohort of adolescents. Therefore, the purpose of this study was to examine the association between RT and MQ in adolescents using readily available data needed to conduct a large-scale cohort study. Our hypothesis was that those who engaged in RT activities two or more times per week would be associated with greater MQ versus those who trained less often (0–1 times per week). Additionally, we also hypothesized that higher frequency RT (≥4 day/week) would be superior to moderate or low frequencies (2–3 day/week) for enhancing MQ. 

## 2. Materials and Methods

### 2.1. Study Population and Design

This study used data from the National Health and Nutrition Examination Survey (NHANES), an ongoing program conducted by the National Center for Health Statistics in the Centers for Disease Control and Prevention [30]. NHANES data for calculating MQ, i.e., muscle strength and mass data, were available only for adolescents (12–15 year) over a two-year period, 2013–2014. 

Of 10,175 individuals that had data reported from this NHANES cycle, 713 were aged 12–15 years old. After excluding those persons who did not have complete data (*n* = 108), a final sample of 605 (318 males and 287 females) was used for the present study. Written consent was previously obtained from NHANES participants and approved by the National Center for Health Statistics Research Ethics Review Board under protocol #98-12. 

### 2.2. Measurement Methods

The exposure variable (independent variable) was identified as RT frequency in which participants were asked about their involvement in RT-based activities performed that were specifically designed to strengthen muscles (e.g., lifting weights, push-ups, sit-ups, etc.), and how frequently they did these activities. No advice was given by NHANES staff on how to appropriately define RT; participants simply reported whether they engaged in any RT activities and how often. For the current study, RT frequency was categorized into the following classification terms: no/none (0 day/week); very low (1 day/week); low (2 day/week); moderate (3 day/week); high (4 day/week); very high (5–7 day/week). 

In a similar manner to a previous analysis done in older adults using dynamometer knee extensor measurements for muscle strength [31], MQ was indirectly calculated based on the available data obtained in NHANES from muscle strength, measured with a handgrip dynamometer, and arm muscle mass, measured using dual-energy X-ray absorptiometry (DXA) (model QDR-4500A, Hologic, Inc., Bedford, MA, USA), which can be found in NHANES Data Documentation files [32]. MQ was quantified by taking measured handgrip strength in kilograms and dividing it by combined arm lean mass excluding bone mineral content in grams.

### 2.3. Participants’ Characteristics

Sociodemographic characteristics were self-reported and included age, sex, race/ethnicity, and household income. The adolescent’s ages ranged from 12 to 15 years old. Race and Hispanic origin were classified as non-Hispanic white, non-Hispanic black, Hispanic, and non-Hispanic other race/ethnicity. Household income was grouped as 0–<$25,000, $25,000–<$45,000, $45,000–<$75,000, and $75,000 or above. The arm fat percentage was the average of the left and right arm percentages obtained from DXA. 

### 2.4. Statistical Analysis

All statistical procedures used in the current study were consistent with appropriate practices and guidelines for medical research [33]. SAS callable SUDAAN 11.0.3 statistical software (Research Triangle Institute, Research Triangle Park, NC, USA) was used for the data analyses. Appropriate sample weights were used in calculating statistically reliable estimates in all analyses given that NHANES data were obtained using a complex, multistage sampling design involving stratification, clustering, and oversampling of specific population subgroups. Standard errors were estimated using Taylor series linearization. Associations for various cohort characteristics with RT frequency and MQ were assessed using chi-squared tests for categorical variables and analysis of variance as well as ordinary least squares regression for continuous variables. Identification of covariates as potential cofounders was based on the significant association of these variables with both the main exposure (frequency of RT) and outcome (MQ). The potential cofounders included in the analyses were gender, race/ethnicity, and arm fat percentage. The association between RT frequency and MQ was calculated by an analysis of covariance. Three different models were produced: (1) unadjusted estimates; (2) adjustments for sex and race/ethnicity; (3) adjustment for model 2 covariates (sex and race/ethnicity) and arm fat percentage. All reported *p*-values were two-sided, and alpha was set to 0.05.

## 3. Results

MQ and RT frequency were associated with sex and arm fat percentage, but not associated with age, race/ethnicity, and household income (Table 1). For our initial analysis, no RT (0 day/week) was compared to 1 day/week and 2–7 day/week RT frequency (Table 2). We found an association between RT for 2–7 day/week and higher MQ versus no exercise at all (0 day/week) in all models (*p*-value for Model 1 = 0.0154; for Model 2 = 0.0030; for Model 3 = 0.0173), whereas this effect was absent for 1 day/week RT. Next, we compared no RT versus 1–2 day/week and 3–7 day/week RT frequency (Table 3) and showed associations between 1–2 day/week and 2–7 day/week RT with greater MQ versus no RT for Model 2 (*p* = 0.0474 and *p* = 0.0081, respectively). However, these associations were only maintained when RT was performed 3–7 day/week in the full model (Model 3, *p* = 0.0382) which is adjusted for arm fat percentage. When no RT was compared with 1–4 day/week and 5–7 day/week RT frequency (Table 4), associations between RT and greater MQ were seen for both frequency ranges in our partially adjusted model (Model 2: *p* = 0.0192 and *p* = 0.0133, respectively). However, our full model demonstrated associations only for RT 5–7 day/week (*p* = 0.0383).

Next, we evaluated no RT versus 1 day/week, 2–3 day/week, and 4–7 day/week RT frequency (Table 5) and found associations in partially adjusted models for both 2–3 day/week and 4–7 day/week (Model 2: *p* = 0.0139 and *p* = 0.0019, respectively). In the full model, an association was maintained only for 4–7 day/week. In contrast, no associations were seen with 1 day/week RT in all models. Finally, no RT was compared to 1 day/week, 2 day/week, 3 day/week, 4 day/week, and 5–7 day/week RT frequency (Table 6). Full models showed associations between RT frequency and MQ compared to no RT for every category (low vs. no RT *p* = 0.0166; high vs. no RT *p* = 0.0294; very high vs. no RT *p* = 0.0358) with the exception of 1 day/week and 3 day/week RT frequency (*p* = 0.4597 and *p* = 0.2451, respectively).

## 4. Discussion

From the present study, the results suggest a minimum RT frequency of 2 day/week is necessary to increase MQ in adolescents. In contrast, when RT only 1 day/week was combined with 2 or more days of RT, these associations were no longer present in our fully adjusted models. Given the results and acknowledging the limitations of our analyses, this work provides similar evidence for the beneficial effects of performing RT twice per week to that characterized previously in older individuals [22,31,34,35,36]. In adults, low RT frequencies promote positive muscle adaptation [27,28]. However, a clear minimal dose of RT in adolescents has not been well established, which would be valuable for those looking to increase strength and/or performance when they are either inexperienced or have limited time to train. [14,27,37]. Our results demonstrated meaningful associations between a RT frequency of two or more days per week and greater MQ, but not when the frequency is reduced to a single session per week. These data contrast to recent evidence showing muscle adaptation can be maintained by performing RT once per week [37,38]. However, these conclusions were specifically made for adults (20–35 years old) and were not based on MQ. Additionally, the focus was on the dose needed to maintain physical performance in those individuals already physically fit and trained, rather than increasing either of these factors, which would likely increase MQ. 

In this study, there were clear statistical associations between RT frequency of 2 or more times per week that were absent when reduced to 1 day per week; therefore, these results demonstrate that the frequency necessary to enhance MQ may be dependent on the age of the individual, as the current analysis in adolescents differs from that of young adults. Given that it has been previously indicated that the adaptive response to exercise is potentially optimized during adolescence, perhaps a slightly higher minimal dose is needed to elicit the responses necessary to promote adaptation and maximize their MQ [16]. Previous literature supports the notion that RT can offer unique benefits for adolescents when appropriately prescribed, such as increased strength, improved neuromuscular performance, improved body composition, augmented bone strength, and a decreased susceptibility to injury [10,12]. Our current results suggest that similarly to older individuals, adolescents are a specialized age cohort that have a higher sensitivity towards the specific exercise prescription presented to them and become more susceptible to the untoward effects of an inappropriate stimulus in comparison to adults. Therefore, an appropriately modified RT prescription is a critical component enabling improved MQ in adolescent individuals, which will ultimately enhance their physical development and performance during this crucial period of maturation and growth. 

RT is the primary means for improving skeletal muscle adaptation [34,39,40]. However, inappropriately prescribed RT can lead to maladaptation, or overtraining, characterized by an absence of muscle mass gains and accompanied by diminished skeletal muscle performance [22,39,41,42]. While traditionally RT interventions have focused on increasing muscle mass, a growing body of evidence indicates MQ is a more sensitive indicator of the capabilities to perform physically demanding tasks and is arguably more functionally relevant than muscle mass [1,2,43]. Therefore, focusing solely on measures of muscle strength, size, or performance is likely to impair our ability to draw meaningful conclusions about the efficacy of a specific training intervention [1]. Therefore, quantifying MQ through a full spectrum approach (e.g., epidemiological, acute/chronic RT intervention trials, animal models, etc.) should be done in adolescent individuals as a biomarker for the optimization of muscle function—which is what this study sought to do using a population-specific prospective experimental design. 

To our knowledge, we are the first to examine comparisons between high or very high RT (4–7 day/week) with lower frequencies (2–3 day/week) in adolescents. Our results showed no differences between RT conducted 2–3 day/week as opposed to 4–7 day/week RT (Table 4), and thus did not support our hypothesis that higher frequency training would be superior to lower frequencies for improving MQ, as both were shown to be generally superior to a very low dose (1 day/week). A previous meta-analysis in adults concluded that higher RT frequency results in greater gains of strength [28]; however, this analysis included young, middle-aged, and older adults, and most of the included studies only compared very low to moderate frequencies (1–3 day/week). In fact, only two studies were included that had high (4 day/week) or very high (5 day/week) comparisons to lower frequencies (1–3 day/week), both of which were reported in untrained young adults [44,45]. Interestingly, there was a significant effect of RT frequency in young adults not present in middle-aged and older adults in which higher RT frequency improved strength [28]. However, a separate meta-analysis in healthy youth (8–18 years old) reported by Chaabene and colleagues [14] found that two sessions per week RT was associated with large effects on improving change-of direction speed, whereas three sessions per week saw only moderate effects, and thus concluded higher frequency training did not appear to be superior for improving change-of-direction speed [14]. Therefore, because leg MQ is an important predictor for improving this characteristic [14,46], the results of this meta-analysis are directly in line with ours indicating that higher frequencies of RT do not elicit further increases in MQ compared to low or moderate frequency training (2–3 day/week). This was confirmed on a follow-up between level comparisons analysis that did not show any statistical significance between lower frequencies and higher frequencies (Appendix A). However, the meta-analysis only included studies where RT was done 1–3 day/week, with none that had high (4 day/week) or very high (5–7 day/week) frequencies [14]. Therefore, direct empirical evidence at these higher doses, in adolescents as well as other age groups, is severely lacking and should be addressed in future research. 

While the results of the present study are important, there were limitations that should be considered when interpreting the results. First, RT activities were based on a self-reported questionnaire administered by NHANES staff without any oversight or supervision as to any of the parameters of their exercise program. In a meta-analysis reported by Lacroix et al. [47], the authors sought to examine the effects of supervised versus unsupervised balance training and/or RT. Studies included in the meta-analysis ranged from 4 to 44 weeks in length, employed a training frequency of 2–5 day/week (with the vast majority being 3 day/week). The types of exercises employed was also mentioned and was comprised primarily of bodyweight and/or barbell exercises of the lower body (e.g., deadlift, squat, lunges, calf raises, and overhead press). The results demonstrated that supervised resistance training induced larger effects on muscle strength and power compared to unsupervised resistance training. Additionally, because these data were self-reported via answers to a questionnaire, there is inherent self-report bias that is intrinsic with this secondary analysis (e.g., social desirability) that may have led to some inaccurate answers [47]. Next, the current analysis was limited to using handgrip strength as a surrogate measure of total body strength. Previous research has shown that handgrip strength is not the best indicator of physical performance [48], and applying a more representative outcome metric for strength, such as a one-repetition maximum or dynamometer peak force, would have been ideal. A similar analysis in older individuals [31] used peak force obtained from a knee dynamometer as a measure of strength used to calculate MQ, but similar data for adolescents was not present in any available NHANES dataset. 

Another limitation was the current analysis was only able to be done in individuals 12–15 years old. The World Health Organization defines adolescence as roughly the period between 10 and 19 years of age [49]; thus, those individuals on the older end of the adolescent spectrum were not included in our study. This was also due to limited data availability in the NHANES datasets, as we only had the capability to calculate MQ for those 12–15 years of age as they were the only ones that had both measures of muscle strength and mass. Additionally, because of the secondary analysis nature of this study design, we do not know how many athletes participated in this survey and therefore cannot report that information; additionally, although the concept of responders versus non-responders to RT has gained attention in recent years [50], we were unable to determine and stratify responders/non-responders for similar reasons as stated above. Finally, based on the purpose of NHANES being a general health assessment of the U.S. population [30], rather than primarily focusing on RT, and the secondary nature of this study, we could not control or account for other parameters of the exercise prescription beyond frequency, such as volume, intensity, and the types of exercise performed. However, the fact that the results of the present study showed an association between enhanced MQ with frequency of training suggests these associations may be even stronger if the exercise prescription was supervised, given the recent findings that supervised exercise leads to larger effects in measures of muscle strength and power [47]. Nonetheless, due to the lack of direct control over the majority of the exercise prescription the interpretation of our findings should be done with caution until they are able to be confirmed in follow-up studies using well-controlled trial studies of RT in adolescents. 

While several limitations have been noted, this study also contained numerous strengths. In regard to the novelty of the current study, to our knowledge, this is the first analysis of NHANES data that has characterized associations between RT frequency and MQ in a large cohort of adolescents. One of the primary strengths of the current study design is its reliance on NHANES. Thus, in comparison to the vast majority of training studies, our current analysis was done with a substantially larger (*n* = 605), more diverse sample size, and from individuals across the entire country, rather than at one specific location, thus eliminating sources of potential bias. Next, our study included data from both male and female participants; as has been noted previously, the majority of RT-related studies have only used male participants and there has been an increasing call for the use of both sexes in this area of research in order to eliminate the inherent bias of results and to improve the applicability of results towards a greater proportion of the population [51]. Additionally, the use of MQ as a main outcome measure of analysis, rather than just strictly muscle strength and/or mass, is an asset given the lack of overall attention and results to this important predictor of function in young populations in previous research [1]. Finally, the presentation of training frequency as an important variable of the exercise prescription is a strength given the fact that this variable has been previously recognized as an often overlooked component of the RT prescription [23].

## 5. Conclusions

In conclusion, our findings in an NHANES population-based cohort of adolescents 12–15 years of age suggest that frequency of RT is an important factor that is associated with greater MQ. Specifically, RT frequency of 2 day/week may be the minimal dose of RT necessary to increase MQ. RT is the most accessible and efficacious exercise intervention to optimize muscle adaptations and subsequently enhance performance necessary to succeed during the execution of physically demanding tasks. As such, our cross-sectional prospective data provides the basis for well-designed, randomized controlled trials that emphasize RT frequency to clearly delineate the mechanisms and confirm the minimal effective dose of RT necessary to enhance the adaptive responses of MQ in adolescents. 

## Figures and Tables

**Table 1 ijerph-19-08099-t001:** Association between characteristics with resistance training and muscle quality (age 12–15 years old).

	Sampled (*n* = 605)	Resistance Training (RT)			Muscle Quality (MQ)	
		No RT (0 Day/Week) *n* = 180	Very Low to Very High RT (1–7 Day/Week) *n* = 425			
		Percent *	Percent *	*p*-Value ^†^	Mean (SE)	*p*-Value ^‡^
Age				0.113		0.1636
12	161	32.8	23.9		12.80 (0.24)	
13	132	19	21.2		12.67 (0.24)	
14	166	30.9	27		13.21 (0.23)	
15	146	17.3	27.8		13.11 (0.21)	
Sex				0.029		<0.0001
Male	318	42.1	55.5		12.22 (0.14)	
Female	287	57.9	44.5		13.75 (0.20)	
Race/Ethnicity				0.66		0.0139
White (Non-Hispanic)	157	54.4	55.4		13.06 (0.20)	
Black (Non-Hispanic)	156	15.2	14.4		12.44 (0.17)	
Hispanic	203	22.9	20.8		12.87 (0.12)	
Others (Non-Hispanic)	89	7.5	9.3		13.43 (0.19)	
Household Income				0.2392		0.1612
0–25 K	159	22.7	17.6		12.83 (0.14)	
25–45 K	143	22.9	18.2		12.63 (0.22)	
45–75 K	94	20.4	20.3		13.10 (0.33)	
75 K+	187	34	43.9		13.15 (0.27)	
Arm Fat Percentage (%)				0.0337		0.0031
<30%	291	33	51.3		12.91 (0.14)	
30–39%	169	33	28.5		13.51 (0.18)	
≥40%	145	34.1	20.1		12.39 (0.19)	

* The percentages and means were weighted. ^†^ *p*-value was from Chi-square test. ^‡^ *p*-value was from ANOVA or regression.

**Table 2 ijerph-19-08099-t002:** Association between resistance training (0 day/week, 1 day/week, and 2–7 day/week) and muscle quality (age 12–15 years old).

	Resistance Training (RT)				
	No RT (0 Day/Week)	Very Low RT (1 Day/Week)	Low to Very High RT (2–7 Day/Week)	*p*-Value ^†^	*p*-Value ^‡^
	(*n* = 180, Est. Pop = 4.3 M)	(*n* = 87, Est. Pop = 2.3 M)	(*n* = 338, Est. Pop = 8.3 M)		
Muscle Quality	Mean (SE)	Mean (SE)			
Model 1	12.68 (0.14)	13.00 (0.42)	13.10 (0.12)	0.4624	0.0154
Model 2	12.54 (0.14)	12.95 (0.38)	13.19 (0.15)	0.3455	0.0030
Model 3	12.64 (0.17)	12.96 (0.35)	13.14 (0.15)	0.4621	0.0173

^†^ *p*-value was from *t*-test: Comparison of muscle quality between no RT and very low RT. ^‡^ *p*-value was from *t*-test: Comparison of muscle quality between no RT and low to very high RT. Model 1 was unadjusted. Model 2 was adjusted with sex and race/ethnicity. Model 3 (=full model) was adjusted with sex, race/ethnicity, and arm fat percentage.

**Table 3 ijerph-19-08099-t003:** Association between resistance training (0 day/week, 1–2 day/week, and 3–7 day/week) and muscle quality (age 12–15 years old).

	Resistance Training (RT)				
	No RT (0 Day/Week)	Very Low/Low RT (1–2 Day/Week)	Moderate to Very High RT (3–7 Day/Week)	*p*-Value ^†^	*p*-Value ^‡^
	(*n* = 180, Est. Pop = 4.3 M)	(*n* = 188, Est. Pop = 4.7 M)	(*n* = 237, Est. Pop = 5.9 M)		
Muscle Quality	Mean (SE)	Mean (SE)			
Model 1	12.68 (0.14)	13.12 (0.22)	13.05 (0.13)	0.0635	0.0632
Model 2	12.54 (0.14)	13.06 (0.22)	13.20 (0.16)	0.0474	0.0081
Model 3	12.64 (0.17)	13.04 (0.20)	13.15 (0.15)	0.1195	0.0382

^†^ *p*-value was from *t*-test: Comparison of muscle quality between no RT and Very Low/Low RT. ^‡^ *p*-value was from *t*-test: Comparison of muscle quality between no RT and moderate to very high RT. Model 1 was unadjusted. Model 2 was adjusted with sex and race/ethnicity. Model 3 (=full model) was adjusted with sex, race/ethnicity, and arm fat percentage.

**Table 4 ijerph-19-08099-t004:** Association between resistance training (0 day/week, 1–4 day/week, and 5–7 day/week) and muscle quality (age 12–15 years old).

	Resistance Training (RT)				
	No RT (0 Day/Week)	Very Low to High RT (1–4 Day/Week)	Very High (5–7 Day/Week)	*p*-Value ^†^	*p*-Value ^‡^
	(*n* = 180, Est. Pop = 4.3 M)	(*n* = 315, Est. Pop = 7.9 M)	(*n* = 110, Est. Pop = 2.7 M)		
Muscle Quality	Mean (SE)	Mean (SE)			
Model 1	12.68 (0.14)	13.12 (0.16)	12.97 (0.27)	0.0402	0.2970
Model 2	12.54 (0.14)	13.11 (0.16)	13.24 (0.23)	0.0192	0.0133
Model 3	12.64 (0.17)	13.06 (0.15)	13.20 (0.23)	0.0787	0.0384

^†^ *p*-value was from *t*-test: Comparison of muscle quality between no RT and low to high RT. ^‡^ *p*-value was from *t*-test: Comparison of muscle quality between no RT and very high RT. Model 1 was unadjusted. Model 2 was adjusted with sex and race/ethnicity. Model 3 (=full model) was adjusted with sex, race/ethnicity, and arm fat percentage.

**Table 5 ijerph-19-08099-t005:** Association between resistance training (0 day/week, 1 day/week, 2–3 day/week, and 4–7 day/week) and muscle quality (age 12–15 years old).

	Resistance Training (RT)						
	No RT (0 Day/Week)	Very Low (1 Day/Week)	Low to Moderate (2–3 Day/Week)	High to Very High (4–7 Day/Week)	*p*-Value ^†^	*p*-Value ^‡^	*p*-Value ^⁋^
	(*n* = 180, Est. Pop = 4.3 M)	(*n* = 87, Est. Pop = 2.3 M)	(*n* = 195, Est. Pop = 4.7 M)	(*n* = 143, Est. Pop = 3.6 M)			
Muscle Quality	Mean (SE)	Mean (SE)	Mean (SE)	Mean (SE)			
Model 1	12.68 (0.14)	13.00 (0.42)	13.19 (0.14)	13.00 (0.18)	0.4624	0.0158	0.1211
Model 2	12.53 (0.14)	12.95 (0.38)	13.12 (0.18)	13.29 (0.16)	0.3438	0.0139	0.0019
Model 3	12.64 (0.17)	12.96 (0.35)	13.06 (0.18)	13.24 (0.16)	0.4600	0.0596	0.0121

^†^ *p*-value was from *t*-test: Comparison of muscle quality between no RT and very low RT. ^‡^ *p*-value was from *t*-test: Comparison of muscle quality between no RT and low to moderate RT. ^⁋^ *p*-value was from *t*-test: Comparison of muscle quality between no RT and high to very high RT. Model 1 was unadjusted. Model 2 was adjusted with sex and race/ethnicity. Model 3 (=full model) was adjusted with sex, race/ethnicity, and arm fat percentage.

**Table 6 ijerph-19-08099-t006:** Association between resistance training (0 day/week, 1 day/week, 2 day/week, 3 day/week, 4 day/week, and 5–7 day/week) and muscle quality (age 12–15 years old).

	Resistance Training (RT)										
	No RT (0 Day/Week)	Very Low (1 Day/Week)	Low (2 Day/Week)	Moderate (3 Day/Week)	High (4 Day/Week)	Very High (5–7 Day/Week)	*p*-Value ^†^	*p*-Value ^‡^	*p*-Value ^⁋^	*p*-Value *	*p*-Value ^
	(*n* = 180, Est. Pop = 4.3 M)	(*n* = 87, Est. Pop = 2.3 M)	(*n* = 101, Est. Pop = 2.4 M)	(*n* = 94, Est. Pop = 2.3 M)	(*n* = 33, Est. Pop = 0.9 M)	(*n* = 110, Est. Pop = 2.7 M)					
Muscle Quality	Mean (SE)	Mean (SE)	Mean (SE)	Mean (SE)	Mean (SE)	Mean (SE)					
Model 1	12.68 (0.14)	13.00 (0.42)	13.23 (0.18)	13.14 (0.21)	13.06 (0.27)	12.97 (0.27)	0.4624	0.0041	0.1302	0.2046	0.2970
Model 2	12.53 (0.14)	12.95 (0.38)	13.16 (0.24)	13.08 (0.23)	13.44 (0.18)	13.24 (0.23)	0.3436	0.0087	0.0891	0.0026	0.0127
Model 3	12.64 (0.17)	12.96 (0.35)	13.11 (0.22)	13.00 (0.23)	13.37 (0.21)	13.20 (0.23)	0.4597	0.0166	0.2451	0.0294	0.0358

^†^ *p*-value was from *t*-test: Comparison of muscle quality between no RT and very low RT. ^‡^ *p*-value was from *t*-test: Comparison of muscle quality between no RT and low to moderate RT. ^⁋^ *p*-value was from *t*-test: Comparison of muscle quality between no RT and moderate RT. * *p*-value was from *t*-test: Comparison of muscle quality between no RT and high RT. ^ *p*-value was from t-test: Comparison of muscle quality between no RT and very high RT. Model 1 was unadjusted. Model 2 was adjusted with sex and race/ethnicity. Model 3 (=full model) was adjusted with sex, race/ethnicity, and arm fat percentage.

## Data Availability

This study used 2013–2014 data from the National Health and Nutrition Examination Survey (NHANES), an ongoing program conducted by the National Center for Health Statistics in the Centers for Disease Control and Prevention [30,32].

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
