# Peer review of "The Relationship between Resistance Training Frequency and Muscle Quality in Adolescents"

_ijerph, 2022, doi:10.3390/ijerph19138099_

Round 1
Reviewer 1 Report
Dear Authors,
This article entitled "The Relationship between Resistance Training Frequency and Muscle Quality in Adolescents" has been read with great pleasure. There are numerous strengths to this study, including its diverse sample and presenting training frequency as an importable variable of exercise prescription. Although the topic is of high interest, some issues of the article require further attention and I will suggest changes to the manuscript.
Introduction. Instead of being too global in the first paragraph, it should be addressed more to adolescents. There should be a more systematic discussion of the definition of muscle quality, as well as a consideration of the adolescents' age range, thus giving more attention to the gender and ethnicity issues in the introduction.
Materials and methods. Could you explain to me why almost 10 years old data is still relevant. Please provide some rationale. As you state in lines 77-79, NHANES staff did not provide any input on how RT should be defined, but have adolescents understood what RT is?
Results. It was rather surprising to see in table 1, the ethnicity part, that non-hispanic adolescents had the highest inactivity rate in RT. Consider discussing both sex and ethnicity results in your discussion.
The arm fat percentage of the adolescents differed from their coevals, so it would be interesting to know how many had been athletes.
Discussion. WHO recommends adults 18-64 years should also perform muscle-strengthening activities at moderate or greater intensity on at least two days a week, as these activities provide additional health benefits. What makes you think that the minimum amount is similar between adolescents and adults? The first paragrpah of the discussion could be stronger.
It has slipped my mind that to RT people are responding differently based on their somatotype - where responders and nonresponders may be found. Can they be distinguished easily?
The fact that you talked about athletes (Ln 196) made me wonder how many of them are in this sample. You might want to consider talking about the general adolescent population instead, if you do not know exactly.
As you talked about your limitations, I wanted also to read about your strengths of this research.
Hope these observations will help you to improve the manuscript.
Author Response
Please see the attachment. References for all citations from our responses to all the reviewers are listed below:
References
- https://www.cdc.gov/nchs/nhanes/about_nhanes.htm. About the National Health and Examination Survey. Available online: (accessed on 03 March 2021).
- Prieske, O.; Chaabene, H.; Moran, J.; Saeterbakken, A.H. Editorial: Adaptations to Advanced Resistance Training Strategies in Youth and Adult Athletes. Front Physiol 2022, 13, 888118, doi:10.3389/fphys.2022.888118.
- Rader, E.P.; Naimo, M.A.; Layner, K.N.; Triscuit, A.M.; Chetlin, R.D.; Ensey, J.; Baker, B.A. Enhancement of skeletal muscle in aged rats following high-intensity stretch-shortening contraction training. Rejuvenation Res 2017, 20, 93-102.
- Stec, M.J.; Thalacker-Mercer, A.; Mayhew, D.L.; Kelly, N.A.; Tuggle, S.C.; Merritt, E.K.; Brown, C.J.; Windham, S.T.; Dell'Italia, L.J.; Bickel, C.S.; et al. Randomized, four-arm, dose-response clinical trial to optimize resistance exercise training for older adults with age-related muscle atrophy. Exp Gerontol 2017, 99, 98-109.
- Thalacker-Mercer, A.; Stec, M.; Cui, X.; Cross, J.; Windham, S.; Bamman, M. Cluster analysis reveals differential transcript profiles associated with resistance training-induced human skeletal muscle hypertrophy. Physiol Genomics 2013, 45, 499-507, doi:10.1152/physiolgenomics.00167.2012.
- Methenitis, S.; Nomikos, T.; Mpampoulis, T.; Kontou, E.; Kiourelli, K.M.; Evangelidou, E.; Papadopoulos, C.; Papadimas, G.; Terzis, G. Different eccentric-based power training volumes improve glycemic, lipidemic profile and body composition of females in a dose-dependent manner: Associations with muscle fibres composition adaptations. Eur J Sport Sci 2022, 1-10, doi:10.1080/17461391.2022.2027024.
- Rader, E.P.; Layner, K.; Triscuit, A.M.; Chetlin, R.D.; Ensey, J.; Baker, B.A. Age-dependent muscle adaptation after chronic stretch-shortening contractions in rats. Aging Dis 2016, 7, 1-13, doi:10.14336/AD.2015.0920.
- Legerlotz, K.; Marzilger, R.; Bohm, S.; Arampatzis, A. Physiological adaptations following resistance training in youth athletes-A narrative review. Pediatr Exerc Sci 2016, 28, 501-520, doi:10.1123/pes.2016-0023.

Reviewer 2 Report
1. There are parts without reference. please fill in all.
Example: line 27-29
2. Please fill out the reliability and validity of the all evaluation tool.
3. Please fill out the inclusion criteria in more detail.
4. Please add more clinical significance to the discussion
Author Response

(The authors gave the same response as above.)

Reviewer 3 Report
I unfortunately must reject this paper. There is inadequate control or description of the independent variable. Obviously this is not a training study, but there is no description of the resistance training activities. Nothing. How does a 12 year old define resistance training? Duration of training? Intensity? Sets, reps etc? Reference #28 - basically the same study with different subjects. The reference is also not updated, it is published, and in the list is "in press".....
Author Response

(The authors gave the same response as above.)

Round 2
Reviewer 1 Report
Even if I was coming to the opinion that an article had serious flaws initially, in the current you have managed to increment the quality of the paper. I may find some really positive aspects that is communicated to the readers. I would say that you manged to solve the main issues in the current submission.
Author Response
Response: We would like to thank the reviewer for the feedback provided during the last round of submission and we are glad that the changes that were made satisfied his or her concerns in full. We agree that the modifications that have improved the overall quality of the current version of the manuscript.
Reviewer 3 Report
Dear authors, thank you for your work and corrections. I have suggested a few changes, please check them out and give me your response. I am not a big fan of data mining but this paper could be valuable if it is used to get the youth more active. Thank you.

Round 3
Reviewer 3 Report
Dear Authors, thank you for your changes. I will approve publication.